# Can Zika Virus Infection in High Risk Pregnant Women Be Differentiated on the Basis of Symptoms?

**DOI:** 10.3390/v12111263

**Published:** 2020-11-05

**Authors:** Nuria Sanchez Clemente, Elizabeth B. Brickley, Marcia Furquim de Almeida, Steven S. Witkin, Saulo Duarte Passos

**Affiliations:** 1Department of Epidemiology, University of São Paulo School of Public Health, São Paulo 01246-904, Brazil; marfural@usp.br; 2Department of Infectious Disease Epidemiology, London School of Hygiene and Tropical Medicine, London WC1E 7HT, UK; Elizabeth.Brickley@lshtm.ac.uk; 3Department of Obstetrics and Gynecology, Weill Cornell Medicine, New York, NY 10065, USA; switkin@med.cornell.edu; 4Jundiaí Medical School, Jundiaí University, Jundiaí, São Paulo 13202-550, Brazil; sauloduarte@uol.com.br

**Keywords:** zika, congenital infections

## Abstract

Zika virus (ZIKV) infection in pregnancy is associated with congenital neurological abnormalities. Our understanding of the full clinical spectrum of ZIKV infection is incomplete. Using data from this prospective cohort study consisting of 650 women attending a high-risk pregnancy clinic during the Zika virus outbreak in Brazil, we investigated the extent to which specific symptoms can be utilized to differentiate ZIKV-infected pregnant women from those with other pregnancy-related problems. All were tested for ZIKV in urine by RT–qPCR. Demographic and clinical data including physical symptoms during follow-up were recorded and analyzed with respect to Zika virus exposure status. Forty-eight (7.4%) women were positive for ZIKV by RT–qPCR. The majority (70.8%) were asymptomatic, and only four ZIKV-positive women (8.3%) reported symptoms during pregnancy that met the WHO case definition. Zika-positive and -negative women reported similar frequencies of ZIKV-like symptoms (as per the WHO definition): fever (16.7% vs. 13.6%), arthralgia/arthritis (10.4% vs. 11.3%), rash (4.2% vs. 5.3%), and conjunctivitis (2.1% vs. 3.2%). Most pregnant women positive for ZIKV in urine are asymptomatic and do not deliver a baby with microcephaly. Physical symptoms alone did not differentiate between high-risk pregnant women positive or negative for ZIKV.

## 1. Introduction

More than three years after the appearance of the Zika virus (ZIKV) in the Americas [1], our understanding of the clinical presentation and consequences of ZIKV infection in pregnant women is still incomplete. The proportion of ZIKV-infected pregnant women who display symptoms, identification of the most common symptoms of ZIKV in pregnancy and their sensitivity and specificity, and whether symptomatic ZIKV-infected pregnant women are at greater risk for delivering an infant with congenital abnormalities remain to be definitively determined.

The incidence of asymptomatic ZIKV infections in pregnant women is widely quoted to be 80% based on estimates from household survey data from Yap Island, Micronesia, 2007 [2]. Systematic reviews to update this figure have used subsequently published epidemiological studies; however, this has proved difficult due to the paucity of studies looking specifically at the clinical spectrum of the ZIKV infection in pregnancy, marked heterogeneity in the reported asymptomatic rates of disease, and small sample sizes [3]. For example, in a recent WHO systematic review looking at the proportion of asymptomatic ZIKV infections in a number of population sub-groups, the proportion of asymptomatic infections in pregnant women varied from 10% to 83% [3]. Data from a countrywide pregnancy cohort study in French Guiana [4] showed striking intra-study heterogeneity in the asymptomatic infection rate of ZIKV-infected pregnant women, related to a number of socio-demographic factors. For example, although 77% of pregnant women overall were asymptomatic, those living in the urbanized coastal areas reported significantly more symptoms than those living in the remote interior (35% versus 17%, *p* = 0.001), and women over 30 were also more likely to report symptoms compared to younger women (28% versus 20%, *p* = 0.03). 

The frequency of individual symptoms reported by pregnant women with confirmed ZIKV infection also varies according to different reports and is hard to assess, because entry criteria for many cohort studies are defined around the presence of being positive for certain symptoms at recruitment, for example, rash or fever [5,6,7,8]. Furthermore, areas that are endemic for ZIKV are also endemic for other flaviviruses; co-infection is possible and complicates the clinical picture [9,10]. The current standard clinical case definition for the ZIKV infection proposed by the World Health Organization (WHO) [11] was last updated in 2016. No ZIKV studies to date have evaluated its sensitivity or specificity in the detection of the clinical ZIKV infection during pregnancy.

There is also no consensus on whether newborns are more likely to exhibit negative sequelae at birth if their ZIKV-infected mothers are symptomatic during pregnancy. A systematic review of 9 case series performed in Brazil and Colombia and 3 cohort studies from the USA concluded that the ratio of symptomatic versus asymptomatic maternal ZIKV infections that resulted in adverse fetal outcomes was 1:1 [12]. However, other publications have hypothesized that women with a symptomatic infection may have a higher viral load than do women with asymptomatic infections and this may translate to a higher probability of birth defects in their offspring [4,13,14].

The aims of the present study are to determine in a region with active ZIKV transmission the relative percentage of ZIKV-positive pregnant women who are asymptomatic and whether symptoms of ZIKV in RT–PCR-positive pregnant women can be differentiated from symptoms present in women with high-risk pregnancies who are ZIKV-negative.

## 2. Materials and Methods

### 2.1. Study Design and Participants

The data reported in this prospective cohort study originated from the Jundiaí Zika Cohort initiated in March 2016 at Jundiaí University Hospital in São Paulo State, Brazil. During the recruitment period (01 March 2016–23 August 2017), all the women attending a high-risk pregnancy clinic, due to the presence of the risk factors threatening the life or health of the pregnant woman or her fetus [15] at any stage of pregnancy, were considered eligible and offered the opportunity to participate in the study. The only exclusion criteria were women with life-threatening conditions or who could not provide informed consent. Research nurses who interviewed the women at enrolment and reviewed their antenatal records gathered detailed demographic, medical, and antenatal information. In addition, all participants were specifically asked if they had experienced the following symptoms during their pregnancy: fever, rash, conjunctivitis, joint pain or swelling, headache, vomiting, lymphadenopathy, bleeding, myalgia, or pruritus. Women who were symptomatic at recruitment, had experienced symptoms earlier in the pregnancy, and/or who developed symptoms consistent with the WHO definition of suspected ZIKV infection (rash and/or fever and at least one of the following symptoms: arthralgia, arthritis, or non-purulent conjunctivitis (see Table 1)) [11] were noted.

Research nurses collected blood, saliva, and urine from all subjects at the time of enrolment, 2–3 weeks after recruitment and subsequently on a 2–3 monthly basis during routine check-ups. Trained volunteers carried out pre-arranged weekly follow-up telephone interviews and inquired whether they had experienced any new symptoms consistent with the ZIKV infection. If symptoms were reported, the women were advised to attend the hospital for clinical review and blood, saliva, and urine were collected at this time as well.

Of note, the methods above describe the recruitment, enrolment, and follow-up of all women in the cohort. At the analysis stage, some individuals or dyads had to be excluded from the analysis, mainly due to missing information pertaining to risk factors or outcomes of interest. This will be described in detail later.

### 2.2. Laboratory Procedures

All laboratory procedures were performed on de-identified samples. It was opted to prioritize the detection of the ZIKV in urine samples by reverse transcriptase–polymerase chain reaction (RT–PCR) and to store blood and saliva for future analysis. Studies have shown that the ZIKV RNA is unlikely to be detected in serum after the first week of illness, whereas it can be detected in urine for at least 2 weeks after the symptom onset [16,17,18].

Total RNA was extracted from urine by a commercial QIAamp Viral RNA Kit (Qiagen^®^, Hilden, Germany) following the manufacturer’s instructions and stored at −80 °C until used. Reverse transcription (RT) and qPCR were performed with a GoTaq^®^ 1-Step RT–qPCR System (Promega^®^, Wisconsin, USA) on an ABI Prism 7500 SDS Real-Time cycler (Applied Biosystems). The primers and probes designed by Lanciotti and colleagues [16] are complementary to the nonstructural 5 protein (polymerase). The RT cycle consisted of a 10 min cycle at 50 °C and a 15 min cycle at 95 °C. The PCR consisted of forty cycles of 15 s at 95 °C and a 1 min cycle at 60 °C. Three positive controls (RNA extracted from positive ZIKV samples) and two negative controls (H_2_O) were included. We considered positive those samples that presented with a threshold cycle (Ct) less than 38.5, as per Lanciotti and colleagues [19]. In the cases where the results were inconclusive, repetitions were performed with serial dilutions.

### 2.3. Neonatal Categorisation

Sonographers specialized in fetal medicine performed antenatal ultrasound scanning at months 3, 5, 7, and 8 in asymptomatic women and monthly in symptomatic women at the São Paulo Radiology Centre using Voluson 730 Expert/Voluson E6, GE equipment. Anthropometric measures at birth (i.e., neonatal weight, length, and head circumference) were obtained for all live-born infants, and the equipment used was consistent. The weight was assessed using digital scales, length—using a recumbent baby length scale, and head circumference—using a standardized non-elastic tape measure. Z-scores for weight, length, and head circumference were determined using the online Intergrowth calculator, which takes into account gestational age and sex [20,21,22]. Gestational age was estimated using the first trimester ultrasound (USS) when available and by the last menstrual period (LMP) when USS was unavailable. Microcephaly was defined as a head circumference z-score of less than −2, determined using the online Intergrowth-21^st^ calculator which takes into account gestational age and sex [21].

### 2.4. Statistical Analysis

The sample size for the cohort was calculated using an estimated prevalence of cases of microcephaly among neonates of ZIKV RT–PCR-positive pregnant women of 2%. A final analytical cohort size of *n* = 531 would give us 80% power to detect a crude relative risk of 2 with a probability of type I error (α) of 5%. Categorical variables were compared among women with and without symptoms, and symptoms were compared between different case definitions using the chi-squared test, except where there were less than 5 in any cell, in which case the Fisher’s exact test was used. For the calculation of sensitivity and specificity of the standard clinical case definition, ZIKV RT–PCR was used as the “gold standard” diagnostic test. All statistical analyses were carried out using the STATA™ version 15.1 software.

### 2.5. Sensitivity and Specificity

The sensitivity and specificity of the current WHO case definition were calculated using the standard formulas, using only the women that had at least one symptom during pregnancy. Asymptomatic individuals were removed for the calculation of sensitivity and specificity. In other words, true positives were defined as symptomatic ZIKV RT–PCR-positive women correctly identified by the WHO case definition and false negatives were defined as symptomatic ZIKV RT–PCR-positive women that did not meet this case definition. True negatives were defined as ZIKV RT–PCR-negative women who were symptomatic but did not meet the full WHO symptomatic case criteria and false positives were defined as symptomatic ZIKV RT–PCR-negative women that met the WHO suspected clinical case criteria. With one caveat, that the current most optimal diagnostic test in areas with active ZIKV transmission is ZIKV RT–PCR and this test has a narrow window for detection of the virus, we cannot assume that any women were truly negative and therefore the number of false positives may be overestimated and the number of true positives underestimated.

## 3. Results

### 3.1. Participant Characteristics

A flow diagram of study participants is shown in Figure 1. A total of 752 women were initially enrolled in the study between March 2016 and August 2017; 26 were eventually lost to follow-up. The women with missing data pertaining to risk factors and outcomes of interest were also excluded. Forty two women did not have a ZIKV RT–PCR sample taken during pregnancy. Additionally, 19 women did not have information on symptoms during pregnancy. Among the remaining 650 women, there were 675 live births (including 25 twin pairs), 17 fetal deaths, and one maternal death.

Of the 650 pregnant women in the final cohort, 32 (4.9%) were recruited in the first trimester, 228 (35.1%) in the second trimester, and 378 (58.1%) in the third trimester. In addition, 48 (7.4%) were ZIKV-positive according to the RT–PCR, 28 (4.3%) were positive by the WHO criteria but were ZIKV-negative, and 574 (88.3%) were negative according to both the RT–PCR and symptoms (Figure 2).

Only 4 (8.3%) of the 48 ZIKV-positive women met the WHO definition of a symptomatic case, 10 (20.8%) had at least one ZIKV-like symptom during pregnancy but did not meet the WHO definition, and 34 (70.8%) did not report any symptoms compatible with the ZIKV infection during pregnancy. Of the 574 pregnant women who did not meet the definition for either a confirmed or a suspected case, 129 (22.3%) had at least one symptom that was consistent with the ZIKV infection and 445 (77.7%) never reported any symptoms during pregnancy.

Socio-demographic characteristics of the study population are shown in Table 2. There were no differences in maternal age, level of education, race, co-habitation, and delivery by cesarean section between symptomatic and asymptomatic women. The time of symptom initiation was known for 13 of the 14 women with the confirmed symptomatic ZIKV infection. Six (46.2%) had symptoms in the first trimester, 4 (30.8%)—in the second trimester, and 3 (23.1%)—in the third trimester. In 25 women who met the WHO case definition but who were ZIKV RT–PCR-negative, 11 (39.3%) had symptoms in the first trimester, 11—in the second trimester and 3 (21.4%) had symptoms in their third trimester. Of note, the mean number of separate urine RT–PCR results available for each woman throughout their pregnancy was 2.02 among ZIKV-positive women and 1.95 among ZIKV-negative women.

Of the 48 women who were ZIKV RT–PCR-positive, 4 (8.3%) reported symptoms during pregnancy that met the WHO definition of a clinical case. Twenty eight (4.3%) ZIKV-negative women also had symptoms consistent with the WHO definition. Among the ZIKV-positive cases, 25 (52.1%) complained of a headache, 8 (16.7%) reported fever, 5 (10.4%) had arthralgia/arthritis, and 2 (4.2%) had a rash. Among the Zika-negative women, 320 (53.2%) complained of headache, 82 (13.6%) had fever, 68 (11.3%) had arthralgia/arthritis, and 32 (5.3%) had a rash during pregnancy. The results are summarized in Table 3. None of these differences reached statistical significance. Of note, the lag time between symptoms and the ZIKV RT–PCR testing ranged from 1 to 177 days (median, 58 days) among the 28 women who had WHO symptoms but tested negative and ranged from 22 to 171 days (median, 53.5 days) among the 4 women who had WHO symptoms and tested RT–PCR-positive.

In our cohort of pregnant women, the sensitivity of the current WHO ZIKV clinical case definition was 28.6% and the specificity was 84.9% (Table 4).

### 3.2. Additional Findings

Although not part of the main objectives of this study, we studied the head circumference of neonates in the cohort at birth as well as the presence of microcephaly as markers of adverse neurological outcomes. Head circumference z-scores of the cohort of newborns at birth were analyzed and stratified by the maternal case definition. The infants of suspected cases, as per the WHO standard case definition, had head circumference z-scores that ranged from −1.05 to 2.71 with a mean of 0.63; and infants of the RT–PCR-confirmed cases had head circumference z-scores that ranged from −2.77 to 2.58 with a mean of 0.51 (*p* = 0.34 using the two-sample t-test). Of the two women with RT–PCR-confirmed ZIKV infection who had newborns with microcephaly (head circumference z-score < −2) at birth, one was asymptomatic and the other reported having had a rash in the first trimester of pregnancy.

## 4. Discussion

In a population of high-risk pregnant women, 7.4% of whom were RT–PCR-positive for ZIKV during pregnancy, most of those positive for the ZIKV infection in urine (70.8%) were asymptomatic. In addition, ZIKV-positive women could not be differentiated from ZIKV-negative women on the basis of symptoms. Lastly, the WHO case definition of suspected ZIKV-positive cases had low sensitivity (28.6%) for detecting RT–PCR confirmed ZIKV cases, but a high specificity (84.9%). Fever was the most frequently reported symptom in ZIKV-infected women (16.7%), but was also reported by 13.6% of women without suspected or confirmed ZIKV infection. The lack of specificity of fever as a symptom for the ZIKV infection has also been reported by others [4,5].

Our results are consistent with a prior report from French Guiana where only 2.4% of Zika-positive women had symptoms that met the WHO standard clinical case definition for the ZIKV disease [4]. The prevalence of asymptomatic ZIKV infections in our cohort (73.9%) is also comparable to reports from Yap Island, Micronesia, during the outbreak in 2007 [16] and French Guiana in 2016 [4] where 81% and 77% of the cohorts were asymptomatic, respectively. In our cohort, rash was infrequently reported in the ZIKV-positive women (4.2%). This is in stark contrast to the high prevalence of rash (90%) among the 31 patients with confirmed ZIKV during the Yap Island outbreak in 2007 [16]. The differences between these two groups include the fact that, in our study, women were pregnant and, therefore, perhaps, differing immunological responses may have led to distinct clinical manifestations [23]. Another factor which has been discussed previously [4] is the difference in skin pigmentation between various study populations which may cause rash to manifest in different ways or be more or less likely to be reported. Underreporting may have also played a part, particularly during the start of the outbreak, as the phenotype of the rash was not well recognized by patients or health workers. Arthritis and arthralgia were present in 10.4% of our confirmed ZIKV-positive women and 11.3% of the women negative for ZIKV infection. This manifestation appears at the top of the list of symptoms in many ZIKV studies [2,4,5,7,24], as does headache [2,4,5,7,8], which, of note, also forms part of the Centers for Disease Control (CDC) ZIKV clinical case definition [25]. In our study, headache was frequently reported by women with confirmed ZIKV-positive infection (52.1%), as well as by ZIKV RT–qPCR-negative women (53.2%). Thus, it may be especially difficult to differentiate between Zika-positive and -negative women on the basis of symptoms when analyzing high-risk pregnancy populations.

Given the low sensitivity of the current WHO clinical case definition for the detection of the clinical ZIKV infection in pregnant women, we propose incorporating arthritis/arthralgia into the major symptoms and adding headache to the minor symptoms. This change would increase the sensitivity to 50% in our study population. Although still far from ideal (as we would still be misclassifying 50% of true positives as false negatives), these additions may be of particular value in pregnant women in regions with active ZIKV transmission where even a presentation with fever and headache (which is otherwise non-specific) would be sufficient to alert a healthcare worker to test for the ZIKV.

Our study has several limitations. It cannot be ruled out that an unknown proportion of women in our cohort who were ZIKV-negative in their urine may, nevertheless, have been positive for this virus. The concomitant use of serology, not available in the present investigation, may have provided additional validation of maternal exposure to this virus, although serological tests for the ZIKV antibody also have significant inherent problems, notably, cross-reactivity with other flaviviruses [26] and the fact that individuals previously exposed to Dengue Virus (DENV) do not mount a ZIKV IgM response [27]. In this study, most women were recruited in the 2^nd^ or 3^rd^ trimester of pregnancy due to time lags, including time to diagnosis of high-risk pregnancy and appointment waiting time. Therefore, women who were infected with the ZIKV in the first trimester of pregnancy might have been missed by the RT–PCR done at recruitment. It is also possible that some women entered the study having previously been exposed to the ZIKV and that some women who tested ZIKV RT–PCR-positive on recruitment were in fact infected prior to pregnancy but still shedding virus in the urine. If this was the case, these women may not have reported symptoms occurring prior to pregnancy and therefore the symptom status may have been incorrectly classified. As prolonged shedding of the ZIKV in the urine is known to occur, there was no limitation set between the onset of symptoms and the ZIKV RT–PCR-positivity. In other words, a woman who was symptomatic but tested RT–PCR-positive in the urine several weeks later was considered to be symptomatic with confirmed ZIKV infection.

As adverse fetal outcomes are fortunately rare following the ZIKV infection, due to small case numbers, it was not possible to make meaningful associations between symptomatology and fetal pathology in this study. This question will have to be revisited in future meta-analyses. Birth outcomes in this cohort were further explored in detail in a separate study [28]. The longitudinal monitoring and developmental follow-up of children in our cohort congenitally exposed to the ZIKV continues. Outcomes of these assessments will be reported in a separate study.

In conclusion, we suggest that institutions revisit clinical case definitions proposed at the start of the ZIKV outbreak and, using available evidence, come to a consensus regarding the most appropriate definition. A more sensitive case definition that includes arthralgia/arthritis as a major symptom and headache as a minor symptom should be considered for pregnant women living in areas with active ZIKV transmission.

## Figures and Tables

**Figure 1 viruses-12-01263-f001:**
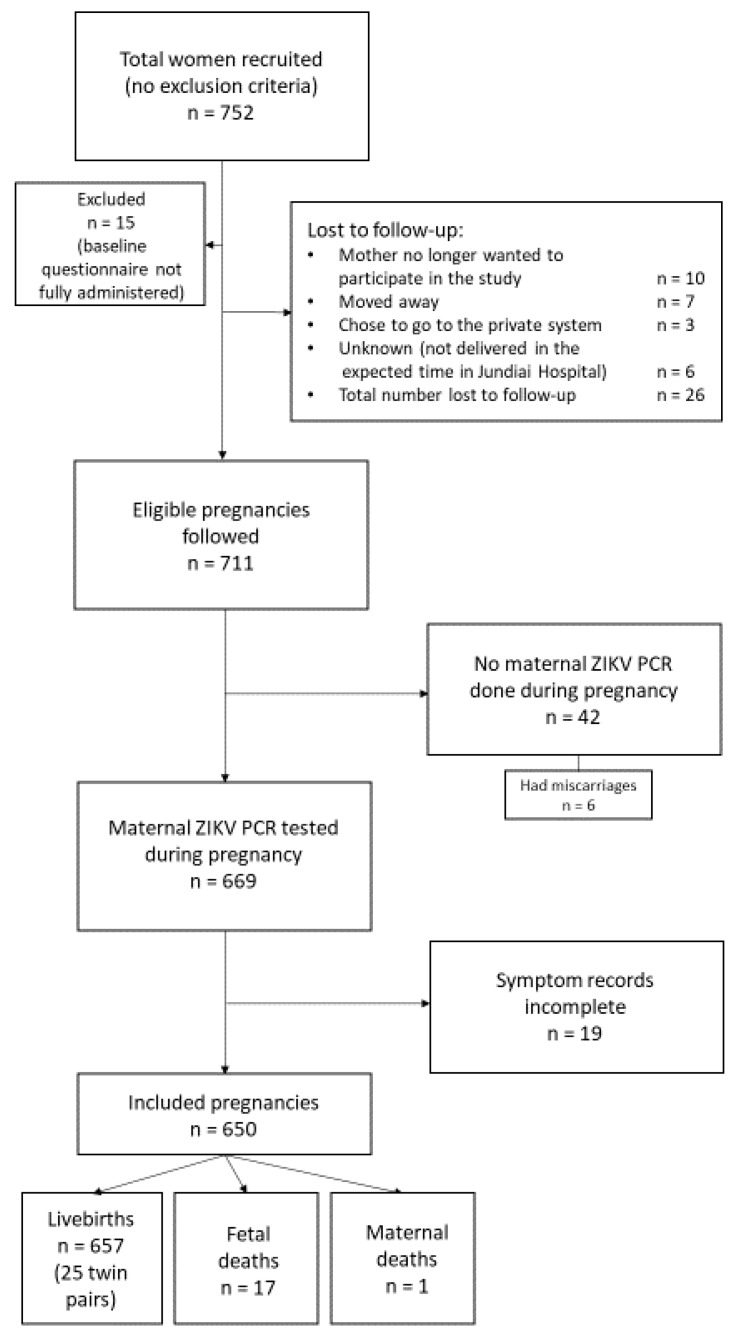
Flow diagram for the selection of study participants.

**Figure 2 viruses-12-01263-f002:**
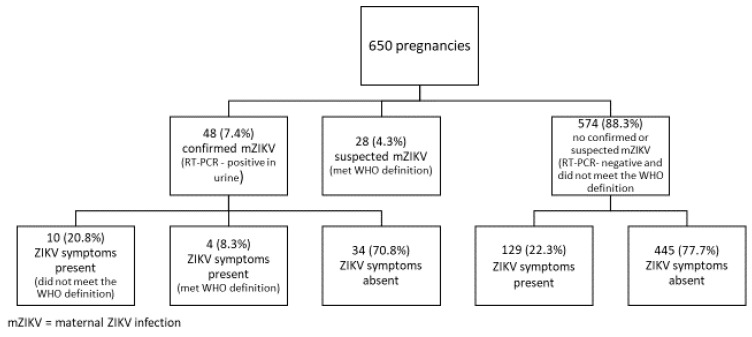
Distribution of pregnant women according to the WHO ZIKV case definitions.

**Table 1 viruses-12-01263-t001:** WHO 2016 interim case definitions for confirmed and suspected Zika virus infection.

Case Definition	Description
Confirmed case	A person with laboratory confirmation of recent Zika virus infection: Presence of Zika virus RNA or antigen in serum or other samples (e.g., saliva, tissues, urine, whole blood); or IgM antibody against ZIKV positive PRNT_90_ for ZIKV with titer ≥ 20 and ZIKV PRNT_90_titer ratio ≥ 4 compared to other flaviviruses; and exclusion of other flaviviruses
Suspected case	A person presenting with rash and/or fever and at least one of the following signs or symptoms: Arthralgia; or Arthritis; or Conjunctivitis (non-purulent/hyperemic)

**Table 2 viruses-12-01263-t002:** Maternal characteristics of participants in the Jundiaí Zika Cohort.

Variable	ZIKV Symptoms Present * (*n* = 171)	ZIKV Symptoms Absent (*n* = 479)	*p*-Value
**Age**			
13–19 years	32 (18.7%)	65 (13.6%)	0.07
20–34 years	112 (65.5%)	305 (63.7%)	
35–46 years	27 (15.8%)	109 (22.8%)	
Missing	0	0	
**Education**			
≤8 years	30 (17.8%)	78 (16.6%)	0.96
9–11 years	40 (23.7%)	106 (22.6%)	
12 years	74 (43.8%)	210 (44.8%)	
>12 years	25 (14.8%)	75 (16.0%)	
Missing	2 (1.2%)	9 (1.9%)	
**Ethnicity/race**			
White	94 (56.3%)	246 (52.5%)	0.86 ^$^
Mixed race	55 (32.9%)	165 (35.2%)	
Black	15 (9.0%)	49 (10.5%)	
Other (Asian/Indigenous)	3 (1.8%)	9 (1.9%)	
Missing	4 (2.3%)	10 (2.1%)	
**Relationship with partner**			
Married/co-habiting	129 (76.3%)	361 (76.7%)	0.93
Single/divorced/widowed	40 (23.7%)	111 (23.4%)	
Missing	2 (1.2%)	8 (1.7%)	
**Type of delivery**			
Vaginal/forceps	75 (51.4%)	200 (50.4%)	0.84
C-section	71 (48.6%)	197 (49.6%)	
Missing	2 (1.4%)	1 (0.25%)	
**ZIKV RT–PCR status**			
Positive in urine	14 (8.2%)	34 (7.1%)	0.64
Negative in urine	157 (91.8%)	445 (92.9)	

* At least one symptom compatible with ZIKV infection as per the current WHO standard clinical case definition. Note: percentages for all categories were calculated with exclusion of those with missing data from the denominator. ^$^ All the *p*-values calculated using the chi-squared test [2] except for those labeled with ^$^, which were calculated using the Fisher’s exact test. The “missing” category was not included as a category when the *p*-value was estimated.

**Table 3 viruses-12-01263-t003:** Symptoms in women with and without PCR-confirmed ZIKV infection.

		No. (%) Positive		
Signs/Symptoms		PCR-Positive *n* = 48	PCR-Negative *n* = 602	*p*-Value **
WHO criteria	Fever	8 (16.7)	82 (13.6)	0.55
	Arthralgia/arthritis	5 (10.4)	68 (11.3)	0.85
	Rash	2 (4.2)	32 (5.3)	0.73
	Conjunctivitis	1 (2.1)	19 (3.2)	0.68
Other symptoms	Myalgia	4 (8.3)	74 (12.3)	0.42
	Headache	25 (52.1)	320 (53.2)	0.89
	Lymphadenopathy	1 (2.1)	33 (5.5)	0.31
Total symptomatic	Fulfilled required WHO criteria *	4 (8.3)	28 (4.7)	0.20
	Did not fulfill WHO criteria Total symptomatic	10 (20.8) 14 (29.2)	129 (21.4) 157 (26.1)	0.20 0.69
No symptoms		34 (70.8)	445 (73.9)	0.64

*A person presenting with rash and/or fever and at least one of the following signs or symptoms: arthralgia, arthritis, or conjunctivitis (non-purulent/hyperemic). ** The chi-squared test was used unless there was < 5 in any cell, in which case the Fisher’s exact test was used.

**Table 4 viruses-12-01263-t004:** Sensitivity, specificity, positive and negative predictive values of the WHO standard clinical case definition for ZIKV infection applied to the symptomatic pregnant women in the Jundiai Zika Cohort.

		Symptomatic Women with the ZIKV Infection		
		ZIKV RT–PCR-Positive	ZIKV RT–PCR-Negative	
Have symptoms that fulfill the WHO standard case definition	Yes	4 (TP)	28 (FP)	PPV = TP / (TP + FP) = 12.5%
	No	10 (FN)	157 (TN)	NPV = TN / (TN + FN) = 94.0%
		Sensitivity = TP / (TP + FN) = 28.6%	Specificity = TN / (FP + TN) = 84.9%	

TP = true positive; TN = true negative; FP = false positive; FN = false negative; PPV = positive predictive value; NPV = negative predictive value.

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
