# Peer review of "Can Zika Virus Infection in High Risk Pregnant Women Be Differentiated on the Basis of Symptoms?"

_viruses, 2020, doi:10.3390/v12111263_

Round 1

Reviewer 1 Report

In this manuscript, the authors are reporting important findings from a prospective cohort study to determine the association of physical symptoms with ZIKV infections in pregnant women. These data are critical to evaluate the extent to which asymptomatic women test positive for ZIKV infections, as well as any correlations among particular symptoms with positive ZIKV test results. The authors determined that most women who test positive are asymptomatic and that physical symptoms are not a reliable indicator of ZIKV infections. These data are critical to improve our understanding of how ZIKV infections are manifest in this population, given the association with microcephaly in some infants born to women who had been infected with ZIKV.

Overall, the study is very clear and well-written and the authors cite previous studies showing the wide range of reported incidence of asymptomatic infections in pregnant women, highlighting the need for surveillance studies from populations of women who are undergoing routine screening.  Although the women in this study were all attending a high-risk pregnancy clinic, which may present some sampling bias relative to the overall population, the data are important and reveal many discrepancies between the presence (or absence) of symptoms that meet the WHO criteria and actual ZIKV infections. A few suggestions to gain more insight into the data presented in the manuscript are outlined below.

  1. Additional information on the stage of pregnancy at enrollment would give a better view of how many women were followed throughout the course of pregnancy or only during later stages.
  2. For the 28 women who reported symptoms but tested negative, an additional breakdown of the timing of symptoms relative to test(s) would be helpful to get a better sense of the possibility of false negatives with regard to testing. If tests were only performed every 2-3 months, it’s not clear how/when the reports of symptoms are related to the time of the tests. Although the weekly telephone interviews should have reduced the testing interval, it’s not clear how many women were tested within a week of reporting symptoms.
  3. For the data on birth outcomes, which was limited to head circumference in the current study, it would be helpful to break it down to see if the timing of infection correlated with the outcome.
  4. Although the birth outcomes are not the primary focus of the study, some discussion of the importance of longitudinal monitoring of infant/child development would be helpful. Even if severe outcomes and microcephaly are rare, there is likely a spectrum of neurodevelopmental effects that could be associated with ZIKV infection that may vary with time of infection and severity of symptoms in the pregnant woman. Being able to associate these data with additional outcome measures would be valuable.
  5. The authors discuss adding arthritis/arthralgia and headaches to the WHO criteria and do show that the percentages of women who tested positive and negative wasn’t different for any individual symptom. However, it would also be helpful to know if any constellation of symptoms was more prevalent in PCR-positive women, regardless of WHO criteria. Do any of the symptoms co-vary in the positive group?

Author Response

Manuscript ID: viruses-949245
Type of manuscript: Article
Title: Can Zika Virus Infection in High Risk Pregnant Women be Differentiated
on the Basis of Symptoms?

Responses to reviewers:

Reviewer 1:

Additional information on the stage of pregnancy at enrolment would give a better view of how many women were followed throughout the course of pregnancy or only during later stages.

Authors’ response: We have added this information in the text.

Pg. 5, lines 163-164: “Of the 650 pregnant women in the final cohort, 32 (4.9%) were recruited in the first trimester, 228 (35.1%) in the second trimester and 378 (58.1%) in the third trimester.”

For the 28 women who reported symptoms but tested negative, an additional breakdown of the timing of symptoms relative to test(s) would be helpful to get a better sense of the possibility of false negatives with regard to testing. If tests were only performed every 2-3 months, it’s not clear how/when the reports of symptoms are related to the time of the tests. Although the weekly telephone interviews should have reduced the testing interval, it’s not clear how many women were tested within a week of reporting symptoms.

Authors’ response: We have provided this information for women who reported symptoms in keeping with the WHO definition and were ZIKV positive and for those who were ZIKV negative. The median time between symptoms and testing was very similar. Because this is a cohort of high risk pregnant women, some women developed complications later in pregnancy and were recruited later. For these women, symptom information was gathered retrospectively therefore resulting in an increased lag time between symptoms and testing.

Pg. 6, lines 189-192: “Of note, the lag time between symptoms and ZIKV RT-PCR testing ranged from 1 to 177 days (median 58 days) among the 28 women who had WHO symptoms but tested negative and ranged from 22 to 171 days (median 53.5 days) among the 4 women who had WHO symptoms and tested RT-PCR positive.”

For the data on birth outcomes, which was limited to head circumference in the current study, it would be helpful to break it down to see if the timing of infection correlated with the outcome.

Authors’ response: Many thanks. There were only 2 cases of microcephaly among ZIKV positive women in our cohort, one these had a rash at 10-12 weeks’ gestation but was only recruited in the 8th month of pregnancy and tested positive then. Interestingly, she continued to test positive in the urine for 5 months after that. The other case was asymptomatic and was also recruited in the 8th month of pregnancy and tested positive at this time. As the authors have published a separate paper on birth outcomes in our cohort, we have added the reference to this manuscript and have suggested that readers refer to this for more details on birth outcomes in our cohort.

Pg 9, lines 286-287: “Birth outcomes in this cohort were further explored in detail in a separate study.28”

Although the birth outcomes are not the primary focus of the study, some discussion of the importance of longitudinal monitoring of infant/child development would be helpful. Even if severe outcomes and microcephaly are rare, there is likely a spectrum of neurodevelopmental effects that could be associated with ZIKV infection that may vary with time of infection and severity of symptoms in the pregnant woman. Being able to associate these data with additional outcome measures would be valuable.

Authors’ response: Thank you. The authors agree, and have added a sentence in the discussion.

Pg 9, lines 287-289: “The longitudinal monitoring and developmental follow-up of children in our cohort congenitally exposed to ZIKV continues. Outcomes of these assessments will be reported in a separate study.”

The authors discuss adding arthritis/arthralgia and headaches to the WHO criteria and do show that the percentages of women who tested positive and negative wasn’t different for any individual symptom. However, it would also be helpful to know if any constellation of symptoms was more prevalent in PCR-positive women, regardless of WHO criteria. Do any of the symptoms co-vary in the positive group?

Authors’ response: Many thanks. Please see table 3 (pg. 7) – none of the symptoms were significantly more prevalent in PCR positive women compared to PCR negative women. Note: this table reports the full constellation of symptoms reported by ZIKV positive women.

Reviewer 2 Report

Zika infection is asymptomatic in 80% of cases. In this sense, a study on clinical symptoms does not seem clinically relevant. However, I understand that being able to discriminate women at risk according to symptoms has a role in places where microbiological study is not available. Can you comment on this?
I would appreciate expanding the discussion regarding the lack of pregnant women with ZIKV infection in the first trimester, since there is evidence that the highest risk of fetal infection occurs when maternal infection occurs in the first trimester.
I would also like the authors to make some reference to fetal deaths and maternal death, is there a suspected causal relationship with Zika?

Author Response

Manuscript ID: viruses-949245
Type of manuscript: Article
Title: Can Zika Virus Infection in High Risk Pregnant Women be Differentiated
on the Basis of Symptoms?

Responses to reviewers:

Reviewer 2:

Zika infection is asymptomatic in 80% of cases. In this sense, a study on clinical symptoms does not seem clinically relevant. However, I understand that being able to discriminate women at risk according to symptoms has a role in places where microbiological study is not available. Can you comment on this?
I would appreciate expanding the discussion regarding the lack of pregnant women with ZIKV infection in the first trimester, since there is evidence that the highest risk of fetal infection occurs when maternal infection occurs in the first trimester.

Authors’ response: We appreciate the reviewer’s observation. As this cohort was made up of high risk pregnancies referred to a tertiary centre, many of them only developed complications later in pregnancy. Even among those who had complications in the first trimester, there was a delay in referral to the hospital, recruitment to the study and carrying out molecular tests.

Please see pg 9, lines 272-278: “In this study, most women were recruited in the 2nd or 3rd trimester of pregnancy due to time lags including time to diagnosis of high-risk pregnancy and appointment waiting time. Therefore, women who were infected with ZIKV in the first trimester of pregnancy might have been missed by RT-PCR done at recruitment. It is also possible that some women entered the study having previously been exposed to ZIKV and that some women who tested ZIKV RT-PCR positive on recruitment were in fact infected prior to pregnancy but still shedding virus in the urine.”

I would also like the authors to make some reference to fetal deaths and maternal death, is there a suspected causal relationship with Zika?

Authors’ response: We thank the reviewer for their comment. We have a separate publication (now referenced in this paper) which looks in detail at fetal outcomes, including fetal death and also mentions one maternal death which occurred in our cohort in a woman who tested positive for Dengue Virus but not ZIKV. However, as both fetal and maternal death are very rare outcomes, we will have to wait for large meta-analyses of cohort data to attempt to answer this question more adequately.

Pg 9, lines 286-287: “Birth outcomes in this cohort were further explored in detail in a separate study.28”
